# Silica-Supported Styrene-Co-Divinylbenzene Pickering Emulsion Polymerization: Tuning Surface Charge and Hydrophobicity by pH and Co-Aid Adsorption

Benoit Fouconnier [1], M. Ali Aboudzadeh [2] and Francisco López-Serrano [3,*]

[1] Facultad de Ciencias Químicas, Universidad Veracruzana, Km 7.5, Avenida Universidad, Col. Santa Isabel, Coatzacoalcos 96538, Mexico; broger@uv.mx
[2] CNRS, Institut des Sciences Analytiques et de Physico-Chimie pour l'Environnement et les Matériaux, University Pau & Pays Adour, E2S UPPA, IPREM, UMR5254, 64000 Pau, France; m.aboudzadeh-barihi@univ-pau.fr
[3] Departamento de Ingeniería Química, Facultad de Química, Universidad Nacional Autónoma de México, Mexico City 04510, Mexico
* Correspondence: lopezserrano@unam.mx

**Abstract:** In this work, polymerizations of styrene (St) in the presence of divinylbenzene (DVB) as a crosslinking agent and sodium 4-vinylbenzenesulfonate (VBS) have been performed in Pickering emulsions, using silica nanoparticles (SNps) as stabilizing agents and ammonium persulfate as a hydrophilic initiator. In oil-in-water Pickering emulsions with alkaline continuous phase (pH = 9) at 1, 2, and 3 wt% DVB (relative to St), polydisperse spheroid copolymer submicronic nanoparticles were obtained. Comparatively, polymerizations performed in Pickering emulsions with acidic continuous phase (pH = 5) allowed preparing St-co-DVB microspheres with core–shell structures at 1 wt% DVB and St-co-DVB hybrid monoliths with bi-continuous morphologies at 2 and 3 wt% DVB. It is noteworthy that this work reports Pickering emulsion polymerization as a new strategy for preparing hybrid percolated scaffolds with bi-continuous porosity. The proposed mechanisms originated by pH, DVB, and VBS and the drastic impact caused on the final morphology obtained, either hybrid particles or monoliths, are discussed herein.

**Keywords:** silica nanoparticles; Pickering emulsion polymerization; microspheres; hybrid monoliths

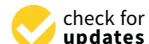



## 1. Introduction

Nowadays, great industrial and academic interests are focused on the design of hybrid nanocomposite materials because they exhibit fascinating properties when inorganic and organic nascent compounds are ingeniously combined [1–5]. These hybrid materials can be engineered in such a way that could bring about electrical, optical, magnetic, or mechanical properties [6–9] while displaying various morphologies, including spherical, raspberry-like, snowman-like, or hollow microspheres [10–13]. Thus, these materials are successfully used in diverse application fields such as electronics, optics, biomedicine, catalysis, cosmetics, adsorption, and separation processes [14–16].

Several methods have been developed to prepare hybrid nanocomposites, such as seed emulsion polymerization, sol-gel preparation, or surfactant-free emulsion polymerizations [17–19]. Nevertheless, Pickering emulsion polymerization has been recognized as a facile route for the synthesis of hybrid materials, as it allows combining intimately two nascent compounds that provide extraordinary properties with the composite materials [20,21].

Contrary to classical emulsions stabilized by surfactants, Pickering emulsions are stabilized by solid nanoparticles, which anchor at the oil–water (O–W) interface, forming a rigid barrier around the dispersed droplets, which makes them highly resistant to coalescence [22,23]. For this reason, Pickering stabilizers have been used to stabilize

high-internal-phase emulsions (HIPE) to prepare highly porous hybrid materials, via polymerization of the continuous phase, with the aim of replacing surfactants, as they can modify the final material properties during their further handling [24,25].

In Pickering emulsion polymerizations, the inorganic nanoparticles do interact with the growing oligomers and these interactions are predominant in the material design since they control the morphology, structure, and properties of the final hybrid nanomaterials [26–28]. Two main polymerization mechanisms have been reported to define the basis of stabilization in Pickering emulsions, which are related to the hydrophobic and hydrophilic nature of the initiator used in the reaction [29]. In the presence of a hydrophilic initiator such as ammonium persulfate (APS) or potassium persulfate (KPS), it has been reported that the homogeneous coagulative mechanism represents the dominating mechanism in Pickering emulsion polymerization, and it leads to the formation of submicronic polymer particles [29–32]. However, when a hydrophobic initiator such as 2,2′-azobis(2-methylpropionitrile) (AIBN) is used, the polymerization is initiated inside the monomer droplets and micrometric latex particles are formed [29,31,33]. However, the mechanisms of polymerization are not so straightforward, and supplementary parameters must be considered, especially the nature of the inorganic nanoparticles, charge, aggregation state, functionalization, and wettability, which can be tuned by adsorption or grafting of organic molecules onto the surface of inorganic nanoparticles [28,34]. These variables are predominant in the polymerization process, as the inorganic nanoparticles directly interfere in the nucleation and coagulation processes, at the early stage of reaction [28,35–37].

During polymerization via the homogeneous coagulative mechanism, the affinity between the inorganic nanoparticles and the polymer matrix is usually provided by modifying the surface of the inorganic nanoparticles with organic functionalizing agents or by means of electrostatic attractions. For instance, Sheibat-Othman et al. [38] reported the polymerization of styrene in the presence of surface-modified silica nanoparticles by a poly(ethylene glycol) monomethyl ether methacrylate (PEGMA)-based macromonomer. The affinity of the inorganic nanoparticles was provided by copolymerization of styrene with the PEGMA chains adsorbed onto the nanoparticles. Lee et al. [39] and Monégier du Sorbier et al. [29] studied styrene polymerization, but the affinity between the polymer and silica nanoparticles was provided by electrostatic interactions.

In another study, Shin et al. [28] performed Pickering emulsion polymerization of styrene stabilized by vinylsilane-functionalized montmorillonite platelets, which resulted in hetero-coagulation nucleation of latex particles. At the early stage of the reaction, they observed fast formation of primary polymer particles by hetero-coagulation, due to attractive interactions between styrene and the surface-functionalized clays.

In contrast, several works have reported that micellar mechanisms can occur, in some experimental conditions, when inorganic nanoparticles' aggregates form pseudo-micelles in the aqueous phase that lead to an increase in the solubility of the hydrophobic monomers [1,29,35,40]. The monomers react with radicals, originated from the hydrophilic initiator thermal decomposition, where the pseudo-micelles provide stability to the oligomers, thus limiting their coagulation. The subsequent primary particles formed are then swollen by monomers to finally form latex particles.

From the above-mentioned examples, it is clear that designing hybrid materials with specific properties requires a good understanding of the mechanistic events that occur during the polymerization. However, suitable modifications or functionalization of the inorganic nanoparticles are usually required to favor the interactions between the inorganic nanoparticles with monomers or the polymer matrix, directly involving the inorganic nanoparticles in the polymerization mechanisms.

Moreover, it is noteworthy that hypercrosslinked polymer microspheres with regular shape and high sorption capacities are required for diverse applications such as high-performance chromatography, ion exchange, hydrogen storage, and water treatment [41,42]. Dispersion polymerization has been reported to be a suitable method for preparing monodisperse micrometric polymer particles through a one-pot reaction, where

all the ingredients are mixed and heated at the initial charge [43]. However, this process becomes troublesome and challenging when preparation of hypercrosslinked microspheres is required [43,44]. Indeed, the presence of a bifunctional monomer as a crosslinking agent modifies the size, morphologies, and size distribution of the final microspheres because it interferes with the nucleation and growth step [43–45]. A number of investigations have pointed out the difficulties in obtaining monodisperse hypercrosslinked particles. Several works have reported some improvements in this situation but without offering a full solution [43].

Similarly, Pickering emulsion polymerization can represent an alternative for preparing hypercrosslinked polymer particles, and some studies have reported the preparation of hypercrosslinked Styrene-Co-Divinylbenzene (St-co-DVB) nanospheres by using surface-modified titania hydrosols [46,47]. Nonetheless, the reactions were performed with less than 4 wt% polymer contents by using a relatively high amount of flocculated titania nanoparticles. The preparation of the St-co-DVB core–shell nanospheres was based on a pseudo-micellar nucleation mechanism.

In this study, St-co-DVB polymerizations were performed in the presence of cationic silica nanoparticles (SNps) and sodium 4-vinylbenzenesulfonate (VBS), as a co-aid and functionalizing agent, in emulsions with basic and acidic water-phase conditions. The objective was to obtain a broad view of the mechanistic events that can occur during the different intervals of Pickering emulsion polymerization, depending on the interactions between SNps with monomers or the polymer. Indeed, based on only one Pickering emulsion formulation, different polymerization mechanisms can occur, depending on the nature of the SNps as Pickering agents. This, in turn, directly depends on the emulsion pH and especially VBS as an SNp-functionalizing agent.

We decided to use positively charged aluminum-coated silica nanoparticles of amphoteric nature, due to the presence of aluminol groups on their surface. This allows tuning the SNps' surface activity by pH changes or by VBS adsorption. The surface activity represents an important variable that directly influences the polymerization processes. At pH = 8.6, bare Ludox CL nanoparticles are highly flocculated [48,49] because the isoelectric point of the SNps is reached at this pH value. Therefore, a limited coagulation process occurs in this situation. However, at pH = 5, VBS monomers are adsorbed onto the SNps, inducing polymerization onto their surface [47]. All polymerization processes are directly governed by the SNps' surface charge and hydrophobicity, also tuned by pH and VBS.

Here, it is shown that St-co-DVB polymerization based on limited homogeneous coagulative nucleation, in silica-supported Pickering oil-in-water emulsions at pH = 9, can represent a facile alternative route to prepare crosslinked particles of submicronic size at a relatively high polymer content (i.e., 20 wt%). Comparatively, submicronic particles are usually obtained in emulsion or microemulsion polymerizations using a relatively high concentration of surfactants. Alternatively, dispersion polymerization can be used as a surfactant-free method, but it gives rise to the formation of micrometric St-co-DVB particles ranging between 1 to 10 micrometers.

Contrarily, at pH = 5 during the polymerization, the affinity between the SNps and the polymers varies not solely as a function of pH but also as a function of VBS functionalization and the crosslinking agent concentration. New polymerization mechanisms have been postulated herein to explain the formation of unusual hybrid percolated monoliths through the formation of bijel in order to exhibit bi-continuous morphology.

## 2. Materials and Methods

### 2.1. Materials

Styrene (St), divinylbenzene (DVB), sodium 4-vinylbenzenesulfonate (VBS), sodium hydroxide (NaOH), acetic acid (AA), and ammonium persulfate (APS) were purchased from Merck KGaA (Mexico City, Mexico). Sodium acetate (SA) was purchased from J. T. Baker (Mexico City, Mexico). Cationic alumina-coated silica nanoparticles (Ludox CL 30, 12 nm of average size, 30 wt% aqueous suspension, 230 $m^2$ $g^{-1}$, pH = 4.5), ammonium

hydroxide, and ammonium chloride were purchased from Sigma-Aldrich S.A. de C.V. (Toluca, Mexico).

### 2.2. Pre-Emulsion Preparation

Pickering emulsions with a dispersed monomer phase mass fraction of 20 wt% were prepared. First, different mixtures of St and DVB containing 1, 2, and 3 wt% of crosslinking agent were prepared and washed three times by using NaOH solution at 10 wt% to remove the inhibitors. Subsequently, these mixtures were washed three times with deionized water and stored under refrigeration before their use. Then, aqueous silica dispersions were prepared by mixing 20 g of Ludox CL and 0.0625 g of VBS in 200 g of water. To perform polymerizations in Pickering emulsions in acidic conditions, 26 g of a pH = 5 buffer solution was added to the Ludox–VBS dispersions, and the resulting solutions were incorporated at the end with the monomer mixtures by using an Ultra Turrax T25 homogenizer (IKA-Werke Works Inc., Wilmington, NC, USA) operating at 15,000 rpm for 10 min. Similarly, for the reactions at alkaline pH, a 1 M NaOH solution was added, drop by drop, to set the pH of the Ludox–VBS dispersions at 9. Then, 26 g of basic buffer solutions were added to the mixtures of the aqueous and oil phases, while stirring at 15,000 rpm for 10 min with the same homogenizer. For emulsions with alkaline continuous phases, the oil-in-water emulsions were first diluted by adding approximately 19 mL of 1 M NaOH solution to fix the pH at 9 before adding the buffer. The composition of the emulsions used in all polymerizations and the buffer solution compositions are summarized in Table 1.

**Table 1.** Composition of the emulsions.

| Emulsion Composition (g) | 1% DVB | 2% DVB | 3% DVB |
|:---:|:---:|:---:|:---:|
| St | 50 | 50 | 50 |
| DVB | 0.5 | 1.02 | 1.54 |
| VBS | 0.0625 | 0.0625 | 0.0625 |
| Ludox CL | 20 | 20 | 20 |
| Water | 200 | 200 | 200 |
| Buffer * | 26 | 26 | 26 |
| APS in 3 mL of water | 1 | 1 | 1 |

* Acid buffer (pH = 5): 3.00 g of $CH_3CO_2Na$ and 8.50 g of $CH_3COOH$ in 88.50 g of water; basic buffer (pH = 9): 0.48 g of $NH_4Cl$ and 1.75 g of $NH_4Cl$ in 97.77 g of water.

The emulsions were placed in a 0.5 L double-walled glass reactor equipped with an external recirculating bath to perform the polymerization at 80 °C. A condenser was fitted in the four-necked reactor vessel, and the pre-emulsions were agitated at 300 rpm by means of a four-bladed steel propeller. Once the temperature of the pre-emulsions reached 80 °C, they were purged with nitrogen for 15 min. Finally, the reactions were initiated by adding 3 mL of water containing 2 wt% APS (with respect to monomers) to the reactor. All polymerization runs were performed in triplicate.

### 2.3. St-Co-DVB Particles and Monoliths' Characterization

The intensity frequency of the St-co-DVB microsphere size distribution prepared in Pickering emulsions at pH = 9 initial charge, containing 1, 2, and 3 wt% DVB in the monomer phase, were measured with a Malvern Zetasizer ZS 90 (Malvern Instruments Ltd., Malvern, Worcestershire, UK) using HCl solutions (pH = 4, refractive index 1.33) to promote the dispersion of the flocculated SNp aggregates. Five DLS measurement runs were performed (3 measurements per run) per reaction.

The core–shell St-co-DVB particles and monolith structures were observed by using a JEOL transmission electron microscope model 2010 (JEOL, Akishima-shi, Japan). The core–shell microsphere latex was dispersed in isopropanol, and then a mixture drop was deposited on a carbon film supported on a copper grid and dried at room temperature

to take TEM images. The monolith samples were dried overnight at 50 °C, cut into slices 0.3 cm in length, and placed onto an aluminum SEM specimen stub for analysis.

## 3. Results and Discussion

### 3.1. SNps Aggregates and Surface-Modified SNps

The bare Ludox CL SNps are characterized by the presence of aluminol groups on their surface, conferring them a positive charge of about +44 mV and 47.8 mV at pH = 4.5 and 3.5, respectively [48,50]. Thus, as soon as VBS was added to the water phase at pH = 5, it adsorbed onto the nanoparticles' surface via its anionic sulfonate group through electrostatic interactions, also inducing a variation in the SNps' hydrophilicity. Therefore, to enhance the affinity of the SNps with growing oligomers during polymerizations, the VBS monomer with a dual functionality was used. On the one hand, it can react with the other monomers, due to its vinyl group, and on the other hand, it adsorbs onto the cationic silica nanoparticles.

However, the aluminol groups of the Ludox CL SNps are amphoteric; therefore, a pH modification alters their charge. Consequently, the polymerization reaction must be performed at a constant pH to avoid any change in the functionality of SNps during the reaction, considering that the initiator thermal decomposition produces hydronium ions. Indeed, as reported in the literature [48,49], the zeta potential value of bare Ludox CL SNps is affected by the pH of the aqueous phase, exhibiting an isoelectric point at pH = 8.6. Therefore, it is concluded that the SNps in the emulsions at pH = 9 are highly flocculated, forming a three-dimensional network [29,51]. Additionally, at this pH, VBS does not interact with the SNps, because both are negatively charged [48,49]. As a result, VBS should be considered as a co-aid or nucleating agent in the polymerizations performed in the emulsions at pH = 9. That being so, the addition of buffers allows keeping the pH almost constant and thus guaranteeing the SNps' aggregated state during the reaction. In such a case, the formation of St-co-DVB microspheres will be promoted through a limited homogeneous coagulative nucleation process, which will be discussed in the next section. The reactions performed at pH values far away from the ones corresponding to the SNps' isoelectric point favor the dispersion of the SNps. Thus, the homogeneous coagulative nucleation would be promoted, leading to the formation of larger microspheres. Similarly, pH = 5 must also be kept constant, on the one hand, to favor a rapid adsorption of VBS onto the SNps' surface and, on the other hand, to avoid the SNps' decomposition and destabilization that could occur if the pH reaches values lower than 3.5 [52].

In our previous work [37], the zeta potential of the VBS surface-modified SNps was measured, and it was observed that the VBS–SNps were weakly flocculated between pH 3.5 and 5. In this pH range, the inorganic particles exhibited a relatively high zeta potential value at almost +30 mV, which was characteristic of a stable colloidal suspension. Consequently, VBS concentration and pH are predominant variables in the polymerization mechanisms that control the morphology, size, and structure of the final copolymer hybrid materials. In the polymerization runs at pH = 5, the SNps are slightly flocculated and are surface-functionalized by the presence of vinyl groups oriented toward the aqueous phase. It is worth noting that the VBS concentration has been optimized to perform polymerization in acid emulsions. The density of the adsorbed VBS anions is 0.132 nm$^{-2}$, which is equal to 60 VBS molecules surrounding one silica nanoparticle.

### 3.2. St-Co-DVB Polymerization in the Presence of Flocculated SNps

The polymerizations in emulsions with alkaline pH yielded the formation of St-co-DVB copolymer microspheres partially covered by SNps. In Figure 1, it can be observed that in all samples, polydisperse submicronic St-co-DVB microspheroidal particles were obtained. It is also clear that the SNps were highly aggregated at pH = 9. The DVB, due to its difference of reactivity with St, interferes in the nucleation and the polymerization growth process and disturbs the polymerization mechanisms, even at low concentration, inducing the production of irregular-shaped latex particles [41,42,44,53]. DVB is more reactive than

St and reacts with VBS, a soluble monomer, in the early stages of the reaction. From DLS measurements and for all runs, as shown in Figure 2, the average polymer particle size fluctuated between 100 and 2000 nm. These results can be attributed to the heterogeneous covering of the polymer particles by the flocculated SNps during the reaction, along with the formation of irregular-shaped polymer particles, such as spheroidal or ellipsoidal, due to the different reactivities of St and DVB. Moreover, the polydispersity index (PDI) of the DLS measurements were 0.269, 0.308, and 0.558 for 1, 2, and 3%, respectively, which indicated heterogeneity of the microsphere dispersions [54].

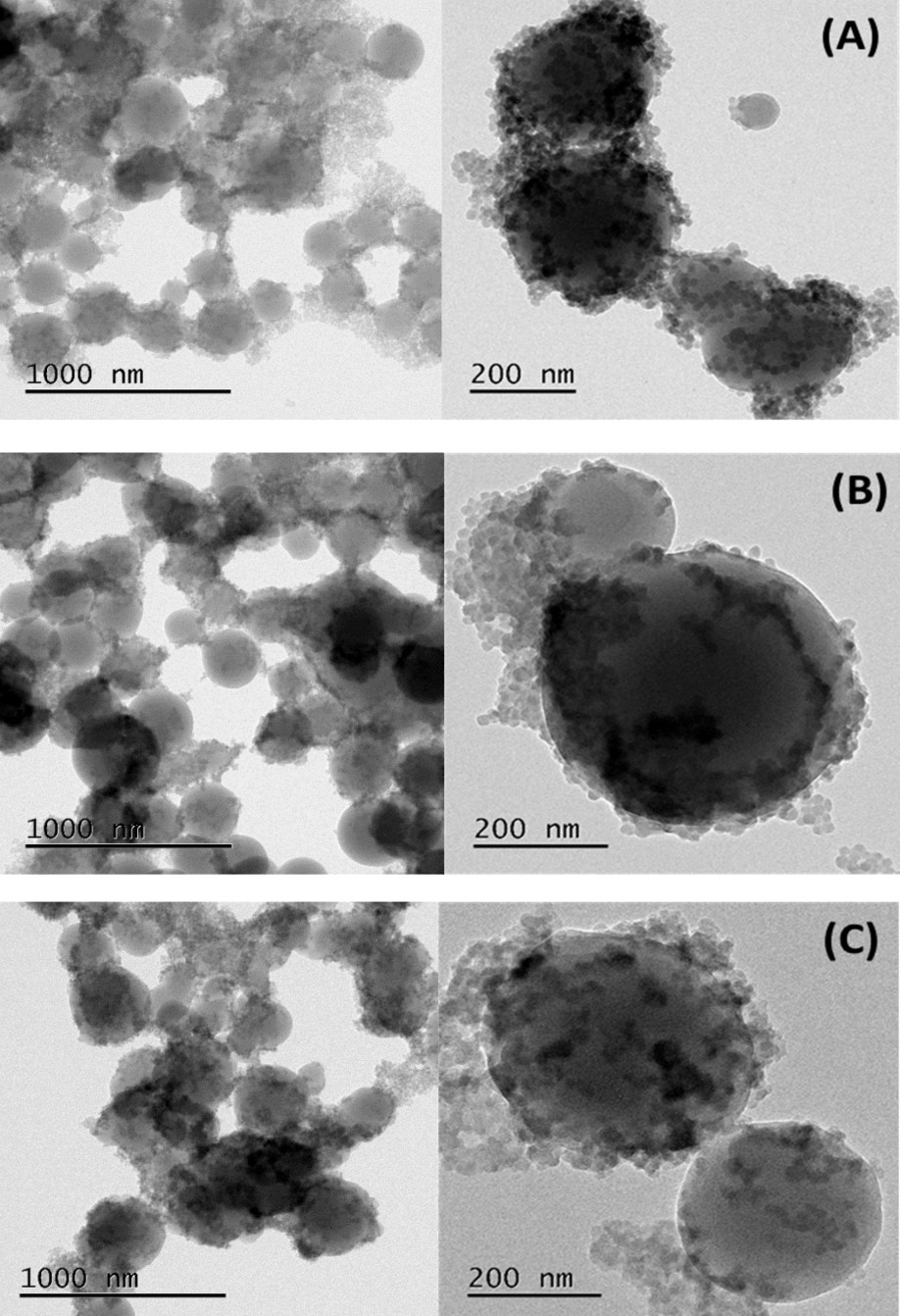

**Figure 1.** TEM images of the St-co-DVB polymer prepared in Pickering emulsion polymerization at pH = 9: (**A**) 1 wt% DVB, (**B**) 2 wt% DVB, and (**C**) 3 wt% DVB.

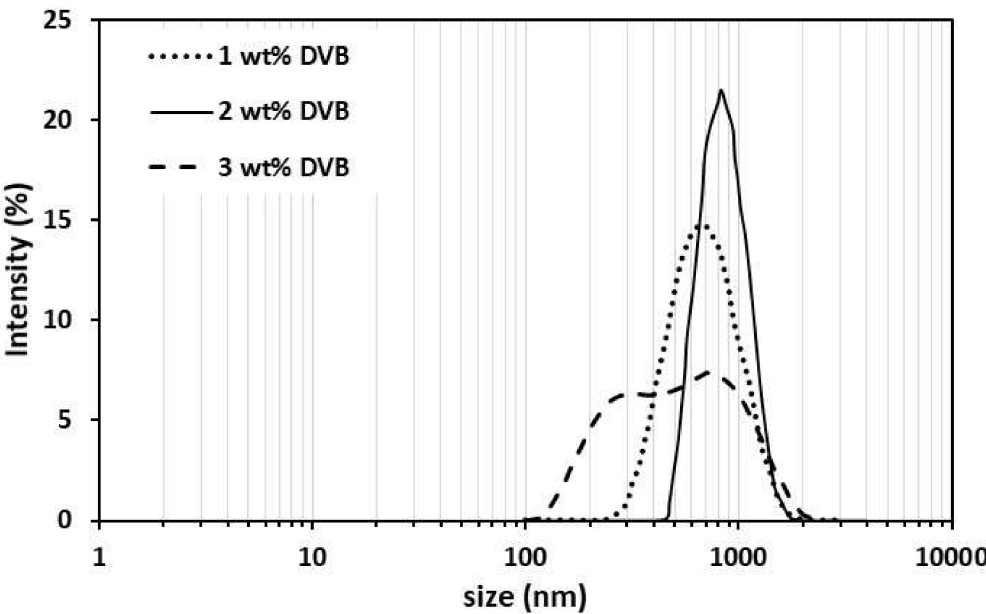

**Figure 2.** Microsphere size distributions prepared in Pickering emulsion polymerization at pH = 9.

Although St-co-DVB polymerization has been extensively studied in dispersion suspension and emulsion polymerizations, few works have been reported on the preparation of copolymer particles of submicron size in Pickering emulsions. Chen et al. [46] reported the preparation of poly(St-DVB) nanospheres with a uniform average size of 100 nm by photocatalytic Pickering emulsion polymerization using titania hydrosols, modified by the adsorption of surfactants. Nevertheless, the monomer dispersed phase content was poor; only 4 wt% monomers (1.05 g St + 0.15 g DVB) were introduced into a 50 mL glass flask containing a high concentration of titania nanoparticles (i.e., approximately 33 wt%, with respect to monomers). The authors proposed a mechanism based on nucleation that occurred inside the surfactant–titania aggregates, which acted as micelles. The nuclei formed inside the aggregates grew via monomer swelling, giving rise to the formation of nanometric mature copolymer particles.

Yin et al. [47] also prepared microspheres of submicrometric sizes, by emulsion polymerization, but their method required large amounts of surfactants. In addition, surfactants are costly and environmentally unfriendly, and they can also affect the properties of the polymeric materials due to their desorption during further polymer handling, so it is preferable to avoid using them. Precipitation polymerization as a surfactant-free method for preparing poly(St-co-DVB) particles represents a good alternative, but this method gives rise to the formation of micron-sized particles [54,55].

In another study, the St Pickering emulsion polymerization at a relatively high polymer content was proposed using a soluble co-aid at low concentration [56]. In these conditions, as soon as the water-soluble initiator is added to the reactor, radicals are formed due to its homolytic decomposition, which will then react with the vinyl groups of VBS monomers, distributed in the three-dimensional SNps' network. As VBS is water soluble, it reacts faster than the low-soluble hydrophobic monomers and therefore promotes the nucleation step by the formation of oligomers. These oligomers will then grow and aggregate to reduce their specific charge area to form precursor polymer particles that will then further grow by monomer swelling. Although SNps do not participate in the polymerization process, as they poorly interact with oligomers of negatively moieties, they indirectly interfere in the nucleation and coagulation processes, due to the formation of a three-dimensional network in the aqueous phase. Indeed, the nanoparticles' aggregates allow reducing considerably the coagulation process, acting as a steric barrier against oligomers' collapse. From these statements, a limited homogeneous coagulative nucleation mechanism is then

postulated [57]. A nucleation step is promoted by the presence of VBS as co-aid occurs, but the collapsing of the growing oligomers is limited, as an SNp's steric barrier impedes direct collapsing, allowing the formation of submicronic polymer particles. Finally, due to the highly flocculated SNps and their poor interactions with the polymer, the latex particles are thus partially covered by the SNps [26,35]. For a better description of the limited homogeneous coagulative nucleation process, a schematic representation of the polymerization mechanisms is presented in Figure 3. In this figure, the flocculated SNps are schematically represented in blue, forming a three-dimensional network, in which monomer droplets are dispersed due to mixing by the propeller. As soon as the initiator is added to the reactor, oligomers (green) form, as a result of the reaction of VBS (dark dash), DVB, and St monomers (red), which will further precipitate to produce primary copolymer particles, which will then grow by monomer swelling, provided by the droplet reservoirs (yellow), to finally give rise to the formation of polydisperse polymer particles with an irregular shape, the result of the difference in reactivities between St and DVB.

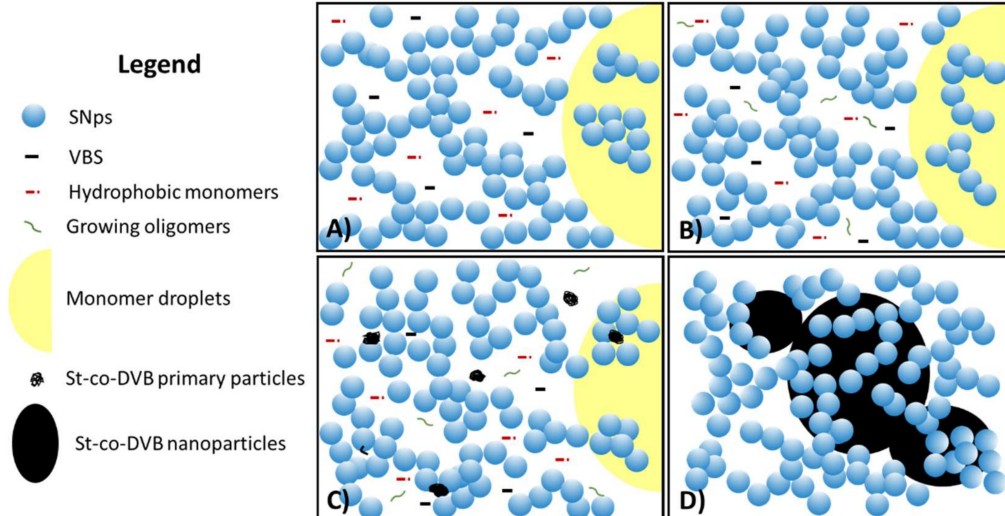

**Figure 3.** Schematic representation of polymerization via limited homogeneous coagulative nucleation. (**A**) Hydrophobic monomers and VBS dissolved in the emulsion aqueous phase where the SNps form a 3D network. (**B**) Oligomer formation in the presence of dissolved monomers. (**C**) Formation of primary polymer particles growing by monomer swelling. (**D**) Polymer microspheres partially covered by SNps.

Herein, an alternative effortless route of poly(St-co-DVB) microsphere preparation is presented at a relatively high polymer particle content at different DVB crosslinking degrees. To obtain monodisperse and homogeneous latex particles, optimizing the copolymerization conditions is required by modifying, for instance, DVB feeding, initiator nature, and concentration and/or by functionalizing the SNps.

### 3.3. St-Co-DVB Polymerization in the Presence of Surface-Modified SNps

When Pickering emulsion polymerizations were performed using VBS surface-modified SNps with 1, 2, and 3 wt% DVB, two scenarios occurred during the polymerization runs. At 1 wt% DVB, copolymer hybrid microspheres with core–shell structures were prepared, as depicted in Figure 4. However, surprisingly, at 2 and 3 wt% DVB, the polymerizations led to the formation of hybrid monolithic structures. The SEM micrographs of these monoliths with bi-continuous morphology are exhibited in Figure 5.

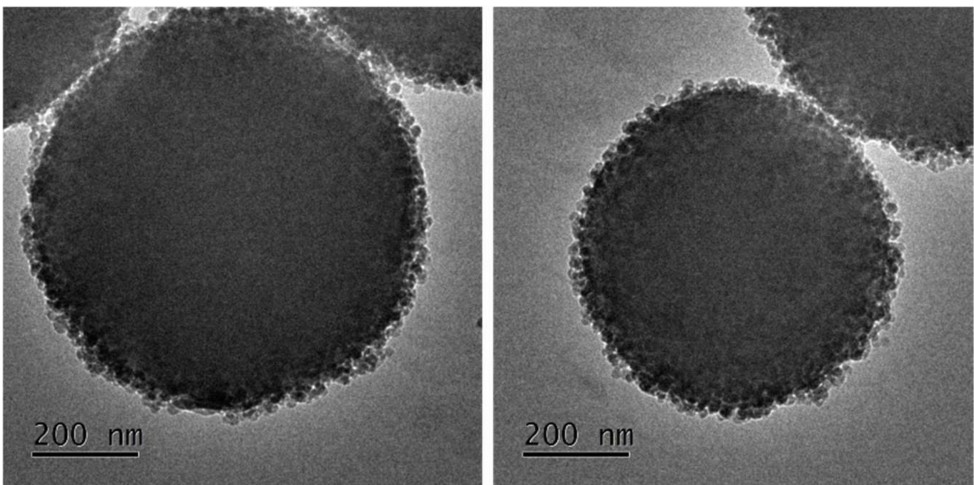

**Figure 4.** TEM images of St-co-DVB microspheres obtained at 1 wt% DVB.

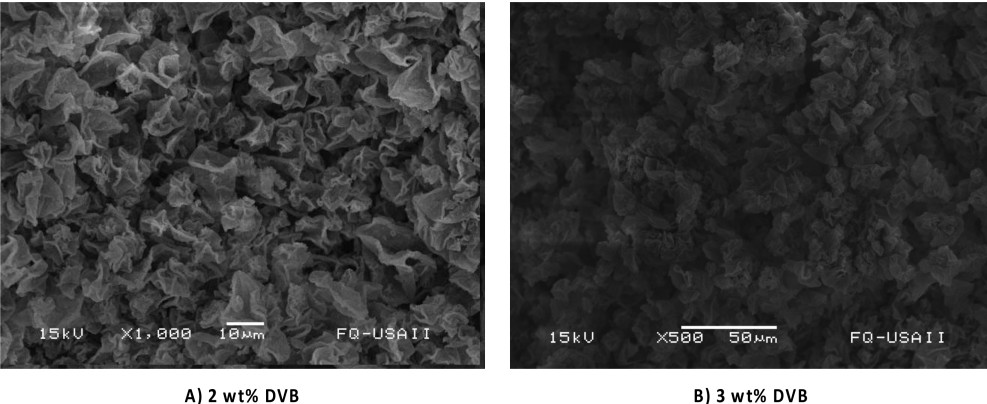

**Figure 5.** SEM images of hybrid monoliths with bi-continuous morphology, (**A**) 2 wt% DVB; (**B**) 3 wt% DVB.

Because of the fact that the emulsion formulations differed solely in the DVB concentration, it has been inferred that the crosslinking agent has a predominant effect on the polymerization mechanisms. It has been shown that during the pre-emulsion preparation process, VBS is added to the silica colloidal suspension in order to enhance its affinity to the polymer matrix and to promote the formation of silica-armored polymer particles [47]. The VBS monomer has a bifunctional nature in our case. On the one hand, it interacts with the SNps' aluminol groups, and on the other hand, due to its vinyl group oriented toward the aqueous phase, it can polymerize [37,47]. Both electrostatic and covalent interactions allow excellent affinity between SNps and the polymer matrix. That being the case, as soon as the reaction is initiated, the free radicals, formed from the homolytic decomposition of APS, can react either with slightly soluble-in-water monomers or with VBS attached onto the SNps' surface to form oligomers. If growing oligomers originate from monomers, their sulfate negative charges, provided by the APS radicals, will ensure their adsorption onto the highly positively charged VBS-SNps [47,58]. Therefore, at the early stage of the polymerization, regardless of the nature of the monomers, it can be considered that the polymerization initiates onto the SNps, forming growing oligomer chains attached onto the SNps' surface. These growing oligomer chains will then reach a critical size, inducing collapsing due to the poor solvent quality of water, thus forming multiple hydrophobic patches onto the SNps [59,60]. These patchy polymer particles are amphiphilic, as they are characterized by a hydrophilic inorganic core, surrounded by hydrophobic pinned micelles. Depending on their hydrophilic–hydrophobic balance, these amphiphilic SNps can aggregate into different anisotropic structures [61,62], which can effectively lead to

the formation of hybrid polymer microspheres or monoliths, during the St-co-DVB polymerization. DVB, as a crosslinking agent, is recognized as interfering in the nucleation and coagulation process. Consequently, during the nucleation process, DVB, due to both its crosslinking character and its concentration effect, can affect the number of growing oligomer chains per SNp and the flexibility and the length of these growing chains [63,64]. Therefore, amphiphilic SNps, with different hydrophilic–hydrophobic characters, can be obtained at the early stage of reactions, which may either self-associate, like micelles in the aqueous phase, or preferentially adsorb, at the monomer droplets' interface, to give rise to microspheres or monolith formation, respectively. The nature and behavior of these amphiphilic SNps, formed in the emulsion aqueous phase during polymerization, led to the preparation of distinct hybrid materials whose mechanisms are described in the following sections.

### 3.3.1. Hybrid St-DVB Microsphere Preparation Using VBS Surface-Modified SNps

As mentioned above, we believe that the reaction is initiated onto the SNps, followed by oligomer chains collapsing onto the SNps, which form hydrophobic patches, also called hemimicelles. By analogy to surfactants, patchy nanoparticles or polymer-grafted nanoparticles can self-assemble into anisotropic structures, depending on the grafting density and size, or chain length, of the polymer chains, relative to the nanoparticle size. Asai et al. [59] showed that grafted polymer nanoparticles behave quantitatively like Janus nanoparticles, which can self-assemble into superstructures. The morphologies for self-assembly can be understood with a geometry-based model of surfactants proposed by Israelachvili [65]. In comparison, Choueri et al. [60] studied surface-patterning nanoparticles with polymer patches and mentioned that polymer chains uniformly grafted onto inorganic nanoparticles can break up into various hydrophobic patches upon solvent quality reduction. They showed that the patch formation and structure were controlled by the polymer length, nanoparticle diameter, and polymer-grafting density. In contrast, Lukach et al. [66] studied the degree of polymerization, bond length, and bond angles of plasmonic polymers. These polymers were prepared by self-assembly of end-tethered gold nano-rods with photoactive macromolecular chains by reducing the solvent quality of end-tethered chains. They photo-crosslinked the polymer chains that resulted in the suppression of the bond-forming ability of the poly(styrene-co-isoprene) chains, a reduction in the inter-nanorods' distance, and an increase in angles between adjacent nanorods. More specifically, the crosslinking effect suppressed the polymer growth and increased the rigidity of copolymer ligands, resulting in a size reduction of the hydrophobic patches at the end of the nano-rods. In addition, crosslinking decreased the effect of the tethered chains' interpenetration, and therefore, weaker hydrophobic interactions between the polymer networks at the nano-rod ends were observed.

In all experiments, due to the low VBS concentration and DVB and St low water-solubility, it can be considered that at the early stage of the reactions, sparsely tethered SNps with St-co-DVB chains are formed [37]. As these chains gain length, the solvent quality decreases, causing agglomeration onto the SNp surface at the end, forming pinned micelles or hydrophobic patches. These hydrophobic zones confer the SNps an amphiphilic character. As microspheres are formed at 1 wt% DVB, it is inferred that the amphiphilic patchy SNps aggregate into copolymer–SNp clusters. Ergo, the St-co-DVB chains can interact and interpenetrate among SNps due to van der Waals interactions. Therefore, self-assembled-like surfactant micelles, exhibiting a polymer hydrophobic core surrounded by SNps, explain this mechanism. This aggregation process leads to the formation of copolymer primary particles, which will then further grow by monomer swelling.

This nucleation mechanism differs from the one postulated by Shin et al. [28] reporting a heterogenous nucleation coagulative process, where the inorganic nanoparticles are incorporated onto the polymer surface during the primary particles' formation, giving rise to the production of clay-armored latex particles. During the early reaction stage, the incorporation of clay platelets into the polymer matrix was promoted, by the silane-

functionalization of the clay particles, resulting in a change in the partition behavior within the Pickering emulsion. In contrast, the nucleation process originates from the in situ formation of tethered SNps, which agglomerate, leading to precursor particles. The interactions between the SNps and the polymer matrix are promoted by the bifunctional VBS monomer character, which allows pulling the SNps from the aqueous phase onto the polymer nanoparticles. This nucleation process, based on the agglomeration of patchy SNps, describes Interval I of the St-co-DVB Pickering emulsion polymerization. Interval II is characterized by the growing of the primary polymer particles until the complete shrinkage of the monomer droplets. Finally, Interval III is characterized by the complete conversion of the monomer inside the polymer particles, without excluding particle nucleation during these last intervals.

### 3.3.2. Hybrid St-DVB Monolith Preparation Using VBS Surface-Modified SNps

At 2 and 3 wt% DVB, the polymerizations led to the formation of monoliths with bicontinuous morphologies, and thus, it can be concluded that copolymer–SNp clusters are not formed at the polymerizations' early stage. As the DVB concentration is higher, in these experiments, the flexibility, size, and number of tethered chains should differ from the tethered chain properties of amphiphilic SNps, formed at 1 wt% DVB. In view of the crosslinking increase, the chain entanglement between tethered SNps is reduced and weaker hydrophobic interactions between amphiphilic particles are induced. At a high crosslinking concentration, it is expected that the polymer tethered chains, attached onto the SNps, are reduced in size and flexibility [66]. Thus, the amphiphilic SNps preferably adsorb at the monomer–water droplets' interface, due to a pronounced amphiphilic character provided by the presence of hard hydrophobic patches on their surface. Therefore, during the ongoing polymerization, hydrophobic patchy SNps progressively anchor at the monomer droplets' interface, where they continue polymerizing, in the presence of hydrophilic surface-modified SNps anchored therein, from the beginning of the pre-emulsion preparation [37]. Consequently, the droplets' interface is characterized by the presence of SNps with rising hydrophobicity, by virtue of their ongoing polymerization and hydrophilic SNps, which will further polymerize. Therefore, the droplets' interface is progressively subjected to a competition between hydrophobic and hydrophilic patchy nanoparticles, both exhibiting an increase in the average molecular weight of the hydrophobic patches caused by polymerization. As a result of the high density of patchy SNps with increasing hydrophobicity at the droplet interface, the radius of curvature of the amphiphilic particles varies continuously until yielding to an emulsion inversion, or separation, because of wettability changes. However, caused by the presence of both hydrophilic and hydrophobic SNps at the oil–water interface, the emulsion destabilization process does not totally succeed, and bijels are formed at this stage of the Pickering emulsion polymerization [37]. Indeed, bijels, or bi-continuous emulsions, are defined as inter-penetrated structures of two immiscible liquids that share a layer of hydrophobic and hydrophilic colloidal nanoparticles at the interface [67–69]. These bijels are defined as jammed emulsion gels that can be prepared via homogenization [70], but their stabilization requires the use of both hydrophilic and hydrophobic nanoparticles, which wet the aqueous and oil phases equally. Shortly after, it is likely that the patchy SNps' adsorption, at the droplet interface, results in the formation of jammed gel emulsions due to the presence of hydrophobic and hydrophilic SNps tethered particles, characterized by polymer chains of different average molecular weight, which allow wetting the water and monomer phases equally [70]. Finally, the ongoing polymerization in the bijels will continue until complete conversion of monomers to generate percolated monoliths with bi-continuous porosity. For better clarity, the polymerization mechanisms that occurred in Pickering emulsions with an acid-continuous-aqueous phase are schematically described in Figure 6.

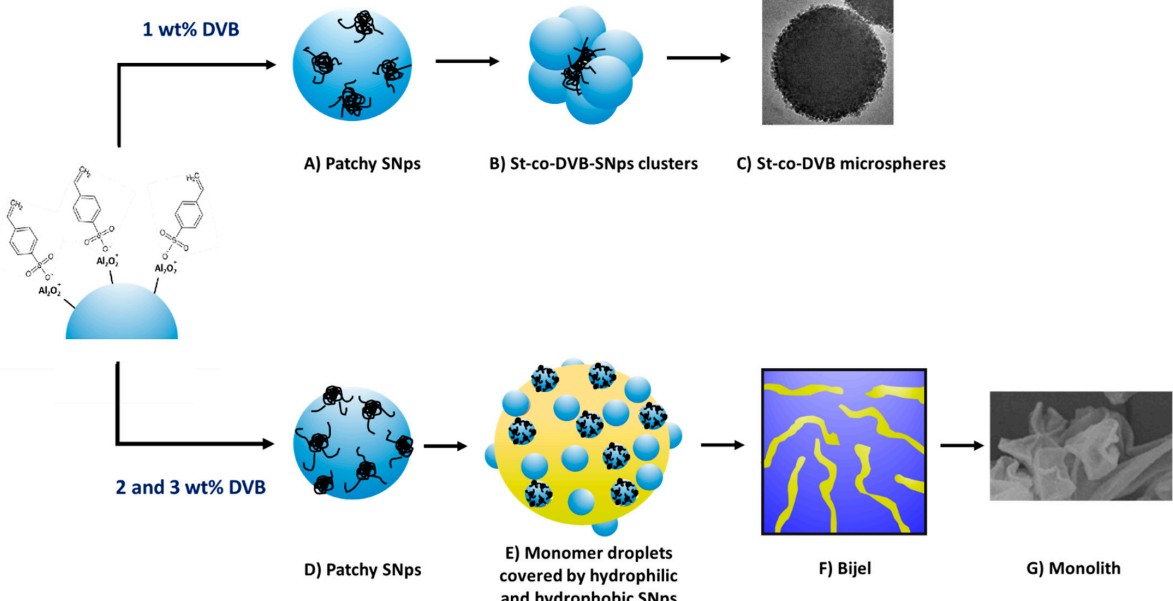

**Figure 6.** Schematic representation of the polymerization mechanisms that occurred in acid Pickering emulsions at (top) 1 wt% DVB and (bottom) 2 and 3 wt% DVB. In the 1 wt% DVB experiment: (**A**) formation of patchy SNps, (**B**) agglomeration of patchy SNps into copolymer–SNp clusters, and (**C**) formation of core–shell copolymer microspheres. In the 2 and 3 wt% DVB experiments: (**D**) formation of patchy SNps, (**E**) adsorption of patchy SNps at the droplets' interface, stabilized by VBS-SNps, (**F**) formation of bijel, and (**G**) formation of monolith due to ongoing bijel polymerization.

Briefly, at pH = 5, DVB as a crosslinker interferes specifically in the nucleation step of the reaction by modifying the length and flexibility of the tethered oligomer chains. By adding different DVB contents in the initial Pickering emulsion charge, it is possible to tune the amphiphilicity of the patchy SNps, which results in the formation of bijels, representing a key step in the formation of percolated monoliths. Thus, this study reveals a new perspective in the use of crosslinking agents to prepare and design new hybrid percolated monoliths with extraordinary properties. Work is in progress to improve the porosity, width, number of folds, rugosity, stiffness, etc., of the final samples by adding higher DVB content or a hydrophilic monomer, such as itaconic acid, in the initial charge. Possible applications of these hybrid materials could be found in the areas of catalysis, contaminants' separation, drug delivery, oil recovery, and semi-conducting materials, just to name a few.

## 4. Conclusions

Polymerizations in emulsion, at different concentrations of DVB, were performed. Varying pH, slightly negatively charged, and cationic VBS surface-modified SNps were obtained. It was clear that the SNps played a predominant role in the mechanisms of Pickering emulsion polymerization. It has been found that the size, structure, and morphology of the final hybrid polymer product are strictly related to the affinity of the SNps to the growing oligomers. When the SNps were highly flocculated (pH = 9), they did not directly participate in the polymerization mechanisms. However, the coagulation process was limited through the formation of a three-dimensional SNp network. Under these conditions, the submicronic St-co-DVB was obtained. Its polydispersity and irregular shape were mainly attributed to the presence of DVB, whose reactivity differs from that of St. On the contrary, when the polymerizations were performed with slightly flocculated VBS surface-modified SNps (pH = 5), the inorganic nanoparticles directly interfered in the polymerization process. VBS adsorbed onto the SNps, allowing the formation of tethered polymer chains onto the SNps. At the early stage of the reactions, amphiphilic SNps were formed via the collapsing of the growing oligomers onto the SNps' surface. Nevertheless,

the flexibility and size of the hydrophobic patches were controlled by the concentration of DVB, which, in turn, affected the hydrophilic–hydrophobic balance of the amphiphilic SNps. At 2 and 3 wt% DVB, the amphiphilic character of the SNps allowed them to be adsorbed at the oil–water droplet interface. Thus, during the polymerizations, the droplet interfaces were characterized by the presence of hydrophilic and hydrophobic SNps where polymerization continued. As the hydrophobicity of the attached amphiphilic SNps increased, due to the ongoing polymerization, their radius of curvature was modified until an emulsion inversion or destabilization was induced, leading to the formation of bijels. The ongoing polymerization of the jammed emulsions, until complete monomer conversion, gave rise at the end to the formation of a hybrid monolith with bi-continuous morphology.

**Author Contributions:** B.F. designed and performed the experiments. F.L.-S. and M.A.A. were involved in planning and result interpretation. B.F. and F.L.-S. wrote the manuscript. B.F., M.A.A. and F.L.-S. reviewed the manuscript. All authors have read and agreed to the published version of the manuscript.

**Funding:** This study was supported financially by CONACyT grant CB-2014-240160. FLS gratefully acknowledges support by UNAM, DGAPA (PAPIIT IN112221).

**Institutional Review Board Statement:** Not applicable.

**Informed Consent Statement:** Not applicable.

**Data Availability Statement:** Data is contained within the article.

**Acknowledgments:** B.F. thanks the Faculty of Chemical Engineering of Universidad Veracruzana.

**Conflicts of Interest:** The authors declare no conflict of interest.

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
