# Peer review of "Silica-Supported Styrene-Co-Divinylbenzene Pickering Emulsion Polymerization: Tuning Surface Charge and Hydrophobicity by pH and Co-Aid Adsorption"

_processes, doi:10.3390/pr9101820_

Round 1

Reviewer 1 Report

In the manuscript by Fouconnier et al. entitled “Synthesis of styrene-co-divinylbenzene hybrid microspheres and percolated monoliths via silica-supported Pickering emulsion polymerization”, the authors describe pickering emulsion polymerizations of styrene (St) in alkaline and acidic conditions, in the presence of divinylbenzene (DVB) as a crosslinking agent and sodium 4-vinyl benzene sulfonate (VBS) as co-aid and functionalizing agent, using silica nanoparticles (SNps) as stabilizing agents and ammonium persulfate as hydrophilic initiator.

In alkaline conditions (pH=9), SNps are highly flocculated; thus, VBS does not interact with them and is considered as a co-aid/nucleating agent. The polymerization in alkaline conditions (pH=9) results in the formation of St-co-DVB copolymer microspheres partially covered by SNps. Overall, an effortless route for the preparation of poly(St-co-DVB) microsphere preparation is presented using different DVB amounts. Further optimization of the copolymerization conditions is required to obtain monodisperse and homogenously covered by SNps, latex particles.

In acidic conditions (pH=5), VBS is adsorbed onto the nanoparticles surface through electrostatic interactions. SNps are surface functionalized in the presence of VBS and slightly flocculated. When Pickering emulsion polymerization is performed using VBS surface functionalized SNps, DVB amount is shown to affect the structure of the final nanocomposites. At 1%wt DVB, copolymer hybrid microspheres with core-shell structures are obtained, whereas at higher DVB amount (2% and 3% wt DVB), hybrid monolithic structures are obtained. The authors describe in detail the formation mechanism of all the produced nanocomposites at the employed conditions. The overall conclusion is that a new method for monoliths preparation has been discovered with work in progress to improve properties of the final structures (e.g. porosity, stiffness etc).

In general, the topic is suitable for the journal. The text is well-structured. The study is interesting but additional clarifications are required before publication. Thus, a major revision is required addressing the comments below:

  1. Novelty of the study presented in the manuscript is questioned. The authors have already published similar studies that they already cite throughout the manuscript (e.g. ref. 37 that is cited in various critical points in the manuscript). The differences with previously published studies are not clearly mentioned. In case the authors aim at providing further details on the mechanism of formation, they should try cite various publications other than their own previously published ones and maybe provide a different title in the manuscript since Ref. 37 (Fouconnier et al., eXPRESS Polymer Letters Vol.15, No.6 (2021) 554–567 accepted in revised form 9 December 2020) has a similar one (Hybrid microspheres and percolated monoliths synthesized via Pickering emulsion co-polymerization stabilized by in situ surface-modified silica nanoparticles). The mechanisms presented in the manuscript for patch particles and monoliths have already been published in Fouconnier et al., eXPRESS Polymer Letters 15, 554 (2021)) (for comparison you can see for example Figure 6 of the present manuscript for the preparation of monoliths vs Figure 8 of the publication in eXPRESS Polymer Letters). Thus, the statement in line 417 “We have discovered a new method for monoliths’ preparation” is misleading since both structure and the associated mechanism have already been published.

Many references are the same in both manuscripts.

Moreover, figure 5A of the present manuscript is the same with figure 7A of Fouconnier et al., eXPRESS Polymer Letters 15, 554 (2021), although conditions in these studies differ only at the amount of SNps (20 g in the present manuscript vs 6 g in eXPRESS Polymer Letters study).

  1. In lines 198-200, the authors state “From DLS measurements and for all runs as shown in Figure 2, the average polymer particle size fluctuated between 100 and 2000 nm”. Could the authors comment on how many runs did they do and how reproducible were their measurements? Maybe error bars in the size distribution measurements shown in Figure 2 could clarify further their statement. Moreover, how reproducible were their nanocomposite synthesis methods? Did they perform the Pickering emulsion polymerizations various times, and did they get same structures every time? This should be clarified in the “Materials and Methods” section.
  2. A refinement of the references list is required.
    1. Refs 43 and 44 (lines 197, 205 and 549-552) seem to be the same.
    2. In line 211, the authors cite Chen et al. as number 44, whereas in page 14, line 553, Chen et al. is ref. 45.  
    3. In line 324, a ref. is missing for the study proposed by Israelachvili.
  3. The Pickering emulsion polymerizations were performed at pH=5 representing acidic conditions and pH=9 representing alkaline conditions. This should be clearly stated throughout the text e.g. lines 188-189, 207 etc. Since only one pH at each range has been tested, the use of “acidic” and “alkaline” conditions terms might be misleading. It would be useful if the authors could clarify if they expect the same results at, for example, pH=3 and pH=11. And what structures do they expect at neutral pH (pH=7)? Since the scope of the manuscript is to present a broad view of the mechanistic events that can occur during the different intervals of the Pickering emulsion polymerization, depending on the interactions between SNps with monomers/polymers, it would be helpful for the readers, even of different backgrounds, to address such issues even with some brief comments.
  4. Could the authors provide a brief justification regarding the use of alumina-coated silica nanoparticles? Why did they not use uncoated silica nanoparticles or a mixture of coated and uncoated particles? A short comment on the importance of surface charge would be helpful for the reader.
  5. In lines 155-156 it is mentioned that “Bare Ludox CL SNps are characterized by the presence of aluminol groups on their surface, conferring them a positive charge of about +44mV at pH 4.5 [37,41].” However, in ref. 37, only the zeta-potential profile of VBS surface modified SNps has been measured and the zeta potential value of the bare Ludox CL SNps is provided by the same reference as used in the present manuscript. Thus, only ref. 41 should be kept in this point.

Some minor points:

  1. The authors should elaborate all abbreviations in text when mentioned for the first time both e.g. O/W, W/O.
  2. The authors should be consistent in keeping or not a gap between the number values and their unit (g. lines 1M in line 133 or 1 M in line 137, 0.0625g in line 128 etc.) and using “=” between pH and its value (e.g. pH = 4.5 in line 119 and pH 9 in line 194 etc.).
  3. Authors should run a grammar and spelling check and correct appropriately some minor misspelling errors. For example:
    1. “hydrophilic” instead of “hydrophylic” in line 308,
    2. “zeta” is missing in front of the word potential for example in lines 167, 174 and 176
    3. “microspheres” instead of “microesferas” in Figure 6
  4. Superscripts should be fixed throughout the manuscript e.g. line 119 and 183-184.
  5. In line 228, the authors state “In other study, …” without having a reference. Do they refer to ref. 49 shown in line 234? An earlier citation of ref. 49 would be helpful.
  6. Lines 422-424 should be deleted as they show instructions for authors as presented in the journal template. In addition, “acknowledgements section” (lines 457-459) should be filled accordingly otherwise deleted, as again instructions for authors as presented in the journal template are shown.

In conclusion, the authors should provide thorough information regarding the reproducibility of the results described. Moreover, they should strengthen the novelty of their study and clarify similarities with their previously published articles. For these reasons, the reviewer recommends that the manuscript should not be published in its present form and a major revision is required. Should further data allow the authors to address the comments provided, the reviewer would be happy to look at the resubmitted manuscript.

Author Response

We appreciate the comments from the Reviewers to help improve the quality and clarity of our Ms. We believe that all the recommendations were satisfactorily followed.

Response to Reviewer 1:

Following Reviewer’s 1 recommendation Tile Now:

Silica-supported styrene-co-divinylbenzene Pickering emulsión polymerization: Tuning surface charge and hydrophobicity by pH and co-aid adsorption

Reviewer 1

We acknowledge the helpful comments offered by this Reviewer, and summarize our detailed responses with corrections, addressed point-by-point below:

Q1. Novelty of the study presented in the manuscript is questioned. The authors have already published similar studies that they already cite throughout the manuscript (e.g. ref. 37 that is cited in various critical points in the manuscript). The differences with previously published studies are not clearly mentioned. In case the authors aim at providing further details on the mechanism of formation, they should try cite various publications other than their own previously published ones and maybe provide a different title in the manuscript since Ref. 37 (Fouconnier et al., eXPRESS Polymer Letters Vol.15, No.6 (2021) 554–567 accepted in revised form 9 December 2020) has a similar one (Hybrid microspheres and percolated monoliths synthesized via Pickering emulsion co-polymerization stabilized by in situ surface-modified silica nanoparticles). The mechanisms presented in the manuscript for patch particles and monoliths have already been published in Fouconnier et al., eXPRESS Polymer Letters 15, 554 (2021)) (for comparison you can see for example Figure 6 of the present manuscript for the preparation of monoliths vs Figure 8 of the publication in eXPRESS Polymer Letters).

R1: This paper deals with the understanding of the mechanisms that can occur in Pickering emulsion, and particularly focuses on the mechanistic events that can occur in a single recipe, depending on the nature of the inorganic nanoparticles, their surface charge and functionalization. In the literature, homogeneous coagulative nucleation, droplets nucleation, micellar nucleation and heterogenous coagulative nucleation have been reported as possible mechanisms that can occur during Pickering emulsion polymerization. However, a suitable balance between the affinity of the inorganic nanoparticles and monomers must be displayed to design hybrid polymers with specific properties. From that perspective, the authors described the use of VBS as co-aid and functionalizing agent of SNps which drastically interfere in the polymerization mechanisms, even at low concentration, resulting in the formation of submicronic polymer particles or unusually percolated monoliths, respectively. It is noteworthy that few works dealing with the preparation of hypercrosslinked St-co-DVB microspheres in Pickering emulsions have been reported in literature. Chen et al. and Yin et al. reported the preparation of monodispersed St-co-DVB nanospheres in Pickering emulsions, but nonetheless, the latex contained less than 4 wt% of solids. In the present work, we studied the polymerization of St in the presence of relatively high DVB concentrations (1, 2, 3 wt%, relative to St), as well as relatively high polymer content.

Accordingly, we compared the results obtained with St-co-DVB microsphere preparation in dispersion polymerization, which is commonly called a surfactant-free method. The techniques for preparation of St-co-DVB hypercrosslinked particles are troublesome and challenging when the DVB amount used is higher than 0.4 wt% DVB. Unlike dispersion polymerization, which results in the formation of micrometric St-co-DVB particles, we prepared submicronic particles in a single step reaction by using highly flocculated SNps to limit the coagulation process. Consequently, Pickering emulsion polymerization can be considered as an alternative to dispersion polymerization for preparing hypercrosslinked St-co-DVB particles.

On the other hand, at pH acid conditions, the SNps were surface functionalized due to the adsorption of VBS, which induced the polymerization onto the SNps. At pH = 5, DVB as a crosslinker perturbs and interferes in the polymerization mechanisms, but specifically modifies the amphiphilicity of the patchy SNps forms at early reaction stages. The heterocoagulation mechanism can lead to the formation of core-shell microspheres (1 wt% DVB) or to the formation of unusual, percolated monoliths via bijel templates.

Action: The introduction was thus modified to strengthen the novelty of this study. We clarified similarities with previously published articles by modifying the introduction as follows (line 97 to 138):

Moreover, it is noteworthy that hypercrosslinked polymer microspheres with regular shape and high sorption capacities are required for diverse applications such as high-performance chromatography, ion exchangers, hydrogen storage, water treatments, etc. [41, 42]. Dispersion polymerization has been reported to be a suitable method for preparing monodisperse micrometric polymer particles through a “one-pot” reaction, where all the ingredients are mixed and heated at the initial charge [43]. However, this process becomes troublesome and challenging when preparation of hypercrosslinked microspheres is required [43, 44]. Indeed, the presence of a bifunctional monomer as crosslinking agent modifies the size, morphologies, and size distribution of the final microspheres because it interferes with the nucleation and growth step [43-45]. A number of investigations have pointed out the difficulties of obtaining monodisperse hypercrosslinked particles. Several works reported some improvements on this situation, but without offering a full solution [43].

Q2. Thus, the statement in line 417 “We have discovered a new method for monoliths’ preparation” is misleading since both structure and the associated mechanism have already been published.

R2: The statement in the original paper (line 417) “We have discovered a new method for monoliths’ preparation” is effectively misleading.

Action: This statement was omitted and a new paragraph was added (above Figure 6) as follows (line 468 to 479):

Briefly, at pH =5, DVB as crosslinker interferes specifically in the nucleation step of the reaction by modifying the length and flexibility of the tether oligomer chains. By adding different DVB contents in the initial Pickering emulsion charge, it is possible to tune the amphiphilicity of the patchy SNps which results in the formation of bijels representing a key step in the formation of percolated monoliths. Consequently, this study reveals a new perspective in the use of crosslinking agents to prepare and design new hybrid percolated monoliths with extraordinary properties. Work is in progress to improve porosity, width, number of folds, rugosity, stiffness and so on, of the final samples by adding higher DVB content or a hydrophilic monomer, such as itaconic acid in the initial charge. Possible applications of these hybrid materials could be found in the areas of catalysis, contaminant´s separation, drug delivery, oil recovery, semi-conducting materials, just to name a few.

Q3. Many references are the same in both manuscripts.

R3: The authors carefully checked the references and 13 new ones (different from the previous manuscript) were added without including our previous works. These new references also strengthen the novelty of our study.

Action: The following references were cited and added as follows:

  1. Šálek, P.; Horák, D. Hypercrosslinked polystyrene microspheres by suspension and dispersion polymerization. e-Polymers. 2011, 11(1), 064.
  2. Tan L.; Tan, B. Hypercrosslinked porous polymer materials: design, synthesis, and applications. Chem. Soc. Rev. 2017, 46(11), 3322–3356.
  3. Peng, B.; Imhof, A. Surface morphology control of cross-linked polymer particles via dispersion polymerization. Soft Matter 2015, 11, 3589–3598.
  4. Šálek, P.; Horák, D.; Hromádková, J. Novel Preparation of Monodisperse Poly(styrene-co-divinylbenzene) Microspheres by Controlled Dispersion Polymerization. Poly. Sci., Ser. B, 2018, 60(1), 9–15.
  5. Castaldo, R.; Gentile, G.; Avella, M.; Carfagna, C.; Ambrogi, V. Microporous Hyper-Crosslinked Polystyrenes and Nanocomposites with High Adsorption Properties: A Review. Polymers 2017, 9, 651.
  6. Kumar, G.; Kakati, A.; Mani, E.; Sangwai, J. S. Nanoparticle Stabilized Solvent-Based Emulsion for Enhanced Heavy Oil Recovery. SPE Canada Heavy Oil Technical Conference. Doi:10.2118/189774-ms.
  7. Li, J.; Stöver, H. D. H. Doubly pH-Responsive Pickering Emulsion. Langmuir 2008 24(23), 13237–13240.
  8. Jia, K.; Guo, Y.; Yu, Y.; Zhang, J.; Yu, L.; Wen, W.; Mai, Y. pH-Responsive Pickering emulsions stabilized solely by surface-inactive nanoparticles via an unconventional stabilization mechanism. Soft Matter, 2021,17, 3346–3357.
  9. Song, X.; Yin, G.; Zhao, Y.; Wang, H.; Du, Q. Effect of an anionic monomer on the pickering emulsion polymerization stabilized by titania hydrosol. J. Polym. Sci. Part A: Polym. Chem., 2009, 47(21), 5728–5736.
  10. Yu, L.; Shi, R.; Qian, H-J; Lu, Z-Y. Versatile fabrication of patchy nanoparticles via patterning of grafted diblock copolymers on NP surface. Phys. Chem. Chem. Phys., 2019, 21, 1417–1427.
  11. Yi, C.; Zhang, S.; Webb, K. T.; Nie, Z. Anisotropic Self-Assembly of Hairy Inorganic Nanoparticles. Accounts of Chemical Research, 2016, 50(1), 12–21.
  12. Cates, M. E.; Clegg, P. S. Bijels: a new class of soft materials. Soft Matter, 2008, 4(11), 2132–2138. doi:10.1039/b807312k 
  13. Di Vitantonio, G.; Wang, T.; Stebe, K. J.; Lee, D. Fabrication and application of bicontinuous interfacially jammed emulsions gels. Appl. Phys. Rev., 2021, 8, 021323.

Q4. Moreover, figure 5A of the present manuscript is the same with figure 7A of Fouconnier et al., eXPRESS Polymer Letters 15, 554 (2021), although conditions in these studies differ only at the amount of SNps (20 g in the present manuscript vs 6 g in eXPRESS Polymer Letters study).

R1: The Reviewer comment is correct and it was taken into account changing the figure in the revised text.

Action: Figure 5A has been replaced by a new image.

Q5. In lines 198-200, the authors state “From DLS measurements and for all runs as shown in Figure 2, the average polymer particle size fluctuated between 100 and 2000 nm”. Could the authors comment on how many runs did they do and how reproducible were their measurements? Maybe error bars in the size distribution measurements shown in Figure 2 could clarify further their statement. Moreover, how reproducible were their nanocomposite synthesis methods? Did they perform the Pickering emulsion polymerizations various times, and did they get same structures every time? This should be clarified in the “Materials and Methods” section.

R5: The review’s comments have been taken into account.

Action: We specified at the end of section 2.2 “Pre-emulsion preparation” the following sentence: All polymerization runs were performed in triplicate (line 183).

We also added section “2.3 St-co-DVB particles and monoliths’ characterization” in the Materials and Methods, as follows (line 184 to 196):

2.3 St-co-DVB particles and monoliths’ characterization

The intensity frequency of St-co-DVB microphere size distribution prepared in Pickering emulsions at pH = 9 initial charge, containing 1, 2 and 3 wt % DVB in the monomer phase, were measured with a Malvern Zetasizer ZS 90 (Malvern Instruments Ltd., Malvern, Worcestershire, UK), using HCl solutions (pH = 4, refractive index 1.33) to promote the dispersion of the flocculated SNp aggregates. Five DLS measurement runs were performed (3 measurements per run) per reaction.

The core-shell St-co-DVB particles and monolith structures were observed by using a JEOL transmission electron microscope model 2010 (JEOL, Japan). The core-shell microsphere latex was dispersed in isopropanol and then a mixture drop was deposited on a carbon film supported on a copper grid and dried at room temperature to take TEM images. Monolith samples were dried overnight at 50 °C, cut into 0.3 cm in length and placed onto an aluminum SEM specimen stub for analysis.

Q6. A refinement of the references list is required.

    1. Refs 43 and 44 (lines 197, 205 and 549-552) seem to be the same.

R6.1. Ref. 43 and 44 are the same and were corrected in the manuscript

Action: ref 43 has been replaced by ref. 53 as followed:

  1. Eren, B.; Solmaz, Y. Preparation and properties of negatively charged styrene acrylic latex particles crosslinked with divinylbenzene. J. Therm. Anal. Calorim. 2019, 141, 1331–1339.
    1. In line 211, the authors cite Chen et al. as number 44, whereas in page 14, line 553, Chen et al. is ref. 45.

R6.2. Ref. was corrected.

Action: Chen et al. is now ref 46.

  1. Chen, Z.; Qin, Z.; Wang, H.; Du, Q. Tailoring surface structure of polymer nanospheres in Pickering emulsion polymerization. J. Colloid Interface Sci. 2013, 401, 80–87.
    1. In line 324, a ref. is missing for the study proposed by Israelachvili.

R6.3. Ref. was added.

Action: Ref. 65 has been added.

  1. Israelachvili, J. N.; Mitchell, D. J.; Ninham, B. W. Theory of self-assembly of hydrocarbon amphiphiles into micelles and bilayers. J. Chem. Soc., Faraday Trans. 2, 1976, 72, 1525-1568.

Q7. The Pickering emulsion polymerizations were performed at pH=5 representing acidic conditions and pH=9 representing alkaline conditions. This should be clearly stated throughout the text e.g. lines 188-189, 207 etc. Since only one pH at each range has been tested, the use of “acidic” and “alkaline” conditions terms might be misleading. It would be useful if the authors could clarify if they expect the same results at, for example, pH=3 and pH=11. And what structures do they expect at neutral pH (pH=7)? Since the scope of the manuscript is to present a broad view of the mechanistic events that can occur during the different intervals of the Pickering emulsion polymerization, depending on the interactions between SNps with monomers/polymers, it would be helpful for the readers, even of different backgrounds, to address such issues even with some brief comments.

R7. As the “acidic” and “alkaline” conditions terms might be misleading, we replaced these terms with their corresponding pH values, respectively.

Experiments were not performed at pH values lower than pH = 4 because at pH values lower than 3.5, the aluminum species on the SNps’ surface can be dissolved, which can promote Ludox CL SNps destabilization [52].

At pH = 9, the SNps are highly flocculated (Ludox CL SNps’ isoelectric point at pH = 8.6). The formation of a three-dimensional network in the aqueous phase, impedes a direct collapsing of the oligomers during reaction. Thus, it is considered that the limiting homogeneous coagulation nucleation is the dominating mechanism during the formation of St-co-DVB microspheres. 

Reactions performed at pH values far away from the one corresponding to the SNps’ isoelectric point may favor the dispersion of the SNps. Thus, the homogeneous coagulative nucleation would be promoted leading to the formation of microspheres of larger size.

Similarly, reactions must be performed at constant pH = 5 to favor on one hand, a rapid adsorption of VBS onto the SNps’ surface, and on the other hand, to avoid the SNps’ decomposition that could occur if pH reaches values lower than 3.5 [52].

Action: The following paragraph has been added in section 3.1 “SNps aggregates and surface modified SNps” (line 223 to 233):

… at pH = 9. Consequently, the addition of buffers allows keeping pH almost constant and thus guaranteeing the SNps’ aggregated state during the reaction. In such case, the formation of St-co-DVB microspheres will be promoted through a limited homogeneous coagulative nucleation process, that will be discussed in the next section. Reactions performed at pH values far away from the corresponding to the SNps’ isoelectric point favors the dispersion of the SNps. Thus, the homogeneous coagulative nucleation would be promoted leading to the formation of microspheres of larger size. Similarly, pH = 5 must be also kept constant to favor on one hand, a rapid adsorption of VBS onto the SNps’ surface, and on the other hand, to avoid the SNps’ decomposition and destabilization that could occur if pH reaches values lower than 3.5 [52].

Q8. Could the authors provide a brief justification regarding the use of alumina-coated silica nanoparticles? Why did they not use uncoated silica nanoparticles or a mixture of coated and uncoated particles? A short comment on the importance of surface charge would be helpful for the reader.

R8 In this work it was decided to use positively charged aluminum-coated silica nanoparticles of amphoteric nature, due to the presence of aluminol groups on their surface. This allows tunning the SNps’ surface activity by pH changes or by VBS adsorption. The surface activity represents an important variable which directly influences the polymerization processes. At pH = 8.6, bare Ludox CL nanoparticles are highly flocculated [48, 49] because the isoelectric point of the SNps is reached at this pH value. Therefore, a limited coagulation process occurs in this situation. On the other hand, at pH = 5, VBS monomers are adsorbed onto the SNps, inducing the polymerization onto their surface [59]. Then, the tethered chains formed onto the SNps will heterocoagulate to form patchy SNps. The hydrophobicity of the patchy nanoparticles depends on DVB content. At 1 wt% DVB, microspheres highly coated by SNps were obtained, whereas at higher DVB concentration, percolated monoliths were prepared through the formation of bijel templates. All these polymerization processes are directly governed by the SNps’ surface charge and hydrophobicity, also tuned by pH and VBS.

Action:  The following paragraph was added ate the end of the introduction (lines 126-135).

It was decided to use positively charged aluminum-coated silica nanoparticles of amphoteric nature, due to the presence of aluminol groups on their surface. This allows tunning the SNps’ surface activity by pH changes or by VBS adsorption. The surface activity represents an important variable which directly influences the polymerization processes. At pH = 8.6, bare Ludox CL nanoparticles are highly flocculated [48, 49] because the isoelectric point of the SNps is reached at this pH value. Therefore, a limited coagulation process occurs in this situation. On the other hand, at pH = 5, VBS monomers are adsorbed onto the SNps, inducing the polymerization onto their surface [47]. All polymerization processes are directly governed by the SNps’ surface charge and hydrophobicity, also tuned by pH and VBS.

Q9. In lines 155-156 it is mentioned that “Bare Ludox CL SNps are characterized by the presence of aluminol groups on their surface, conferring them a positive charge of about +44mV at pH 4.5 [37,41].” However, in ref. 37, only the zeta-potential profile of VBS surface modified SNps has been measured and the zeta potential value of the bare Ludox CL SNps is provided by the same reference as used in the present manuscript. Thus, only ref. 41 should be kept in this point.

R9. The authors delated ref. 37 and added ref. 50.

Action: Ref. 50 has been added in the references.

  1. Kumar, G.; Kakati, A.; Mani, E.; Sangwai, J. S. Nanoparticle Stabilized Solvent-Based Emulsion for Enhanced Heavy Oil Recovery. SPE Canada Heavy Oil Technical Conference.

Some minor points:

  1. The authors should elaborate all abbreviations in text when mentioned for the first time both e.g. O/W, W/O.

Action:  In the manuscript, “O/W Pickering emulsions” were replaced by “oil-in-water Pickering emulsions”. Oil/water (O/W) interface was specified in line 45 for the first time and was used like that in the manuscript.

  1. The authors should be consistent in keeping or not a gap between the number values and their unit (g. lines 1M in line 133 or 1 M in line 137, 0.0625g in line 128 etc.) and using “=” between pH and its value (e.g. pH = 4.5 in line 119 and pH 9 in line 194 etc.).

Action: All these inconsistencies were corrected in the manuscript.

  1. Authors should run a grammar and spelling check and correct appropriately some minor misspelling errors. For example:
    1. “hydrophilic” instead of “hydrophylic” in line 308,
    2. “zeta” is missing in front of the word potential for example in lines 167, 174 and 176
    3. “microspheres” instead of “microesferas” in Figure 6

Action: Grammar and spelling errors were checked and the manuscript has been revised by an English native spoken person.

  1. Superscripts should be fixed throughout the manuscript e.g. line 119 and 183-184.

Action: Superscripts have been fixed in the manuscript.

  1. In line 228, the authors state “In other study, …” without having a reference. Do they refer to ref. 49 shown in line 234? An earlier citation of ref. 49 would be helpful.

Action: The authors refer to ref. 56 and has been cited earlier in the paragraph (line 289).

  1. Lines 422-424 should be deleted as they show instructions for authors as presented in the journal template. In addition, “acknowledgements section” (lines 457-459) should be filled accordingly otherwise deleted, as again instructions for authors as presented in the journal template are shown.

Action: This paragraph has been deleted.

Reviewer 2 Report

López-Serrano and co-workers reported the synthesis of styrene-co-divinylbenzene microspheres and monoliths via silica-supported Pickering emulsion polymerization. However, this manuscript offers no significant improvement from their work published in eXPRESS Polymer Letters (EXPRESS Polym. Lett. 2021, 15(6), 554-567). In fact, this manuscript is an exact rewrite of the original manuscript and is not advisable to publish.

Author Response

We acknowledge the helpful comments offered by this Reviewer, and summarize our detailed responses with corrections, addressed point-by-point below:

Reviewer 2:

Q1. López-Serrano and co-workers reported the synthesis of styrene-co-divinylbenzene microspheres and monoliths via silica-supported Pickering emulsion polymerization. However, this manuscript offers no significant improvement from their work published in eXPRESS Polymer Letters (EXPRESS Polym. Lett. 2021, 15(6), 554-567). In fact, this manuscript is an exact rewrite of the original manuscript and is not advisable to publish.

R1: The authors consider that this manuscript is not a rewritten version of our previously-published work in eXPRESS Polymer Letters (EXPRESS Polym. Lett. 2021, 15(6), 554-567). Actually, this work is an extended study of the published paper. Indeed, based on only one Pickering emulsion formulation, different polymerization mechanisms can occur, depending on the nature of the Silica nanoparticles, as Pickering agents, which in turn directly depends on the emulsion pH and especially VBS as a silica nanoparticles' functionalizing agent. At basic pH, VBS absorbs onto the silica nanoparticles and homogeneous coagulative nucleation is promoted. On the contrary, at acid pH, VBS does absorb onto the silica nanoparticles, promoting the in-situ formation of amphiphilic silica nanoparticles, due to heterogeneous coagulative nucleation. These mechanisms were described in detail in this manuscript emphasizing the fact that pH and VBS concentration are substantial variables that govern the polymerization mechanisms, giving rise to either classical polymer microspheres or unusual materials, such as percolated monoliths.

The preparation of polymer monoliths via Pickering emulsion polymerization is a new synthesis process and we therefore consider that it deserves diffusion in high-standard journals such as Processes.

Action 1: The introduction was thus modified to strengthen the novelty of this study. We clarified similarities with previously published articles by modifying the introduction as follows (line 97 to 148):

Moreover, it is noteworthy that hypercrosslinked polymer microspheres with regular shape and high sorption capacities are required for diverse applications such as high-performance chromatography, ion exchangers, hydrogen storage, water treatments, etc. [41, 42]. Dispersion polymerization has been reported to be a suitable method for preparing monodisperse micrometric polymer particles through a “one-pot” reaction, where all the ingredients are mixed and heated at the initial charge [43]. However, this process becomes troublesome and challenging when preparation of hypercrosslinked microspheres is required [43, 44]. Indeed, the presence of a bifunctional monomer as crosslinking agent modifies the size, morphologies, and size distribution of the final microspheres because it interferes with the nucleation and growth step [43-45]. A number of investigations have pointed out the difficulties of obtaining monodisperse hypercrosslinked particles. Several works reported some improvements on this situation, but without offering a full solution [43].

Action 2: The following references were added and discussed along the Ms. to support the new findings of this work:

  1. Šálek, P.; Horák, D. Hypercrosslinked polystyrene microspheres by suspension and dispersion polymerization. e-Polymers. 2011, 11(1), 064.
  2. Tan L.; Tan, B. Hypercrosslinked porous polymer materials: design, synthesis, and applications. Chem. Soc. Rev. 2017, 46(11), 3322–3356.
  3. Peng, B.; Imhof, A. Surface morphology control of cross-linked polymer particles via dispersion polymerization. Soft Matter 2015, 11, 3589–3598.
  4. Šálek, P.; Horák, D.; Hromádková, J. Novel Preparation of Monodisperse Poly(styrene-co-divinylbenzene) Microspheres by Controlled Dispersion Polymerization. Poly. Sci., Ser. B, 2018, 60(1), 9–15.
  5. Castaldo, R.; Gentile, G.; Avella, M.; Carfagna, C.; Ambrogi, V. Microporous Hyper-Crosslinked Polystyrenes and Nanocomposites with High Adsorption Properties: A Review. Polymers 2017, 9, 651.

  1. Kumar, G.; Kakati, A.; Mani, E.; Sangwai, J. S. Nanoparticle Stabilized Solvent-Based Emulsion for Enhanced Heavy Oil Recovery. SPE Canada Heavy Oil Technical Conference. Doi:10.2118/189774-ms.
  2. Li, J.; Stöver, H. D. H. Doubly pH-Responsive Pickering Emulsion. Langmuir 2008 24(23), 13237–13240.
  3. Jia, K.; Guo, Y.; Yu, Y.; Zhang, J.; Yu, L.; Wen, W.; Mai, Y. pH-Responsive Pickering emulsions stabilized solely by surface-inactive nanoparticles via an unconventional stabilization mechanism. Soft Matter, 2021,17, 3346–3357.

  1. Song, X.; Yin, G.; Zhao, Y.; Wang, H.; Du, Q. Effect of an anionic monomer on the pickering emulsion polymerization stabilized by titania hydrosol. J. Polym. Sci. Part A: Polym. Chem., 2009, 47(21), 5728–5736.

  1. Yu, L.; Shi, R.; Qian, H-J; Lu, Z-Y. Versatile fabrication of patchy nanoparticles via patterning of grafted diblock copolymers on NP surface. Phys. Chem. Chem. Phys., 2019, 21, 1417–1427.

  1. Yi, C.; Zhang, S.; Webb, K. T.; Nie, Z. Anisotropic Self-Assembly of Hairy Inorganic Nanoparticles. Accounts of Chemical Research, 2016, 50(1), 12–21.

  1. Cates, M. E.; Clegg, P. S. Bijels: a new class of soft materials. Soft Matter, 2008, 4(11), 2132–2138. doi:10.1039/b807312k 
  2. Di Vitantonio, G.; Wang, T.; Stebe, K. J.; Lee, D. Fabrication and application of bicontinuous interfacially jammed emulsions gels. Appl. Phys. Rev., 2021, 8, 021323.

The title has been changed to:

Silica-supported styrene-co-divinylbenzene Pickering emulsión polymerization: Tuning surface charge and hydrophobicity by pH and co-aid adsorption

Reviewer 3 Report

This manuscript presented by Francisco López-Serrano is a nice piece of work and a new breakthrough. This work shows the synthesis of styrene-co-divinylbenzene hybrid microspheres and percolated monoliths via silica-supported Pickering emulsion polymerization at different concentrations of divinylbenzene. For different experimental phenomena, different explanations and mechanisms have been proposed in detail, which is benefit for a good understanding of mechanistic events that occur in Pickering emulsion polymerization. To my opinion, this manuscript deserves to be published in “Processes” after minor revision.

  1. If we continue to improve the content of DVB, what is about to happen for the final morphology? The similar morphology in this work or some new or interesting phenomenon.
  2. For the whole manuscript, there are some of mistakes in the format of references. Therefore, please check the whole manuscript carefully before publication.

Author Response

We acknowledge the helpful comments offered by this Reviewer, and summarize our detailed responses with corrections, addressed point-by-point below:

Reviewer 3:

Q1: If we continue to improve the content of DVB, what is about to happen for the final morphology? The similar morphology in this work or some new or interesting phenomenon.

R1: Reactions were performed in Pickering emulsions at pH = 5 by adding higher DVB content (i.e., 5% 6% wt% DVB relative to St). Similar percolated monoliths were obtained and the effect of DVB content upon the specific surface area and material brittleness is being undertaken in our laboratory at the moment.

Reactions at higher DVB content have not yet been performed at pH = 9.

Action: This following paragraph has been added to the final of section 3.3.2. (line 468 to 479):

Briefly, at pH =5, DVB as crosslinker interferes specifically in the nucleation step of the reaction by modifying the length and flexibility of the tether oligomer chains. By adding different DVB contents in the initial Pickering emulsion charge, it is possible to tune the amphiphilicity of the patchy SNps which results in the formation of bijels representing a key step in the formation of percolated monoliths. Consequently, this study reveals a new perspective in the use of crosslinking agents to prepare and design new hybrid percolated monoliths with extraordinary properties. Work is in progress to improve porosity, width, number of folds, rugosity, stiffness and so on, of the final samples by adding higher DVB content or a hydrophilic monomer, such as itaconic acid in the initial charge. Possible applications of these hybrid materials could be found in the areas of catalysis, contaminant´s separation, drug delivery, oil recovery, semi-conducting materials, just to name a few.

Q2. For the whole manuscript, there are some of mistakes in the format of references. Therefore, please check the whole manuscript carefully before publication.

Action: The manuscript has been carefully checked.

Round 2

Reviewer 2 Report

I agree with the authors explanation of their work as an extended study to their original study (EXPRESS Polym. Lett. 2021, 15(6), 554-567), and I do realize that the preparation of polymer monoliths via Pickering emulsion polymerization is a new synthesis process and detailed mechanistic investigation is needed to better understand and advance this type of polymerization. I commend the authors response to the reviewers comments and I think the process needs more mechanistic understanding and this manuscript is a step in the right direction. I would recommend the manuscript to be published after minor corrections to the language and few spelling mistakes. 

Author Response

In the final revision the spelling and grammar mistakes were corrected. These are written in blue.

We appreciate the comments and suggestions performed by this Reviewer.

Regards

Francisco López-Serrano